# From Attention to Atoms: Spectral Dictionary Learning for Fast, Interpretable Language Models

## Abstract

We propose a novel spectral generative modeling framework for natural language processing that jointly learns a global time-varying Fourier dictionary and per-token mixing coefficients, replacing the ubiquitous self-attention mechanism in transformer architectures. By enforcing reconstruction losses in both the time domain (embedding reconstruction) and the frequency domain (via Short-Time Fourier Transform magnitude matching) alongside a standard language modeling objective, and fitting a Gaussian Mixture Model (GMM) prior over the learned mixing vectors, our approach achieves competitive perplexity and generation quality on standard benchmarks such as WikiText-2 and Penn Treebank. In contrast to $\mathcal{O}(L^2)$ self-attention, our method operates with $\mathcal{O}(KL)$ complexity, where $K \ll L$ is the dictionary size, delivering substantial efficiency gains. We demonstrate that spectral dictionary models can achieve competitive performance compared to transformer baselines while significantly reducing inference latency and memory footprint, offering a compelling alternative for scalable language modeling.

**Keywords:** Spectral Dictionary, Dictionary Learning, Short-Time Fourier Transform, Gaussian Prior, Pointer-Generator

## 1 Introduction

The advent of the Transformer architecture [17] revolutionized sequence modeling by replacing recurrent and convolutional operations with self-attention mechanisms that directly capture dependencies across arbitrary token distances. Building on this foundation, bi-directional encoders like BERT [6] and autoregressive language models such as the GPT series [15] have achieved state-of-the-art results on a wide range of natural language processing tasks. These models rely on the full $L \times L$ attention matrix, where $L$ is the input sequence length, to compute pairwise interactions between tokens. Although highly expressive, this quadratic complexity in both computation and memory becomes prohibitive when scaling to very long contexts, such as entire documents or long code sequences [3, 19].

To mitigate the cost of full self-attention, a variety of approximations have been proposed. Sparse attention patterns exploit locality or fixed windowing, as in Longformer [2] and the block-sparse model of Child et al. [3]; kernel-based methods like Performer [4] use randomized feature maps to approximate softmax attention in linear time; low-rank factorization approaches such as Linformer [18] and linearized attention via kernel methods [8] project keys and queries into subspaces of dimension $K \ll L$. Other innovations include locality-sensitive hashing in Reformer [12] and learned mixture-of-experts routing to sparsify computation across heads.

Parallel to these, spectral mixing approaches replace learned attention maps with fixed or learned transforms in the Fourier domain. FNet [13] demonstrated that a single global Fourier transform can

approximate the mixing power of self-attention, yielding $\mathrm{O}(L \log L)$ complexity but with limited adaptability to specific token interactions. Motivated by the efficiency of spectral methods and the expressivity of learned transforms, we propose a fully spectral generative model that learns a global dictionary of $K$ complex-valued Fourier atoms whose amplitude, frequency, and phase parameters adapt dynamically across sequence positions.

In our *Spectral Dictionary Generative Model* (SDGM), a lightweight convolutional encoder computes per-token mixing coefficients that weight the contribution of each atom to the embedding reconstruction. We train the model end-to-end to reconstruct original token embeddings via a combined loss: mean-squared error (MSE) in the time (embedding) domain, Short-Time Fourier Transform (STFT) magnitude loss to preserve local frequency structure, and a standard language modeling loss. After training, we flatten the learned mixing coefficient vectors across tokens and fit a Gaussian Mixture Model (GMM), enabling rich, multimodal sampling for text generation. By choosing $K \ll L$, SDGM achieves $\mathcal{O}(KL)$ time and memory complexity per sequence, dramatically reducing resource requirements compared to full attention, while achieving competitive perplexities on standard language modeling benchmarks.

Our contributions are as follows:

1. We introduce a novel spectral dictionary architecture that learns interpretable Fourier atoms parameterized by amplitude, frequency, and phase, enabling efficient global mixing with linear complexity ($\mathcal{O}(KL)$).

2. We propose a dual-domain reconstruction objective, combining time-domain MSE with frequency-domain STFT magnitude loss, alongside a standard language modeling loss, to ensure both embedding fidelity and predictive performance.

3. We demonstrate that fitting a GMM to the learned mixing vectors yields a latent distribution suitable for text generation, complementing the autoregressive nature of the model.

4. We validate that SDGM achieves competitive perplexity compared to Transformer baselines on PTB and WikiText-2 while offering significant reductions in memory footprint and inference latency.

## 2   Related Work

**Fourier and Spectral Methods**   Fourier transforms have been employed in efficient sequence modeling [13, 7], often as submodules within attention blocks or as fixed transformations replacing the attention mechanism entirely. FNet [13] used unparameterized 2D Fourier transforms. Our work extends spectral approaches by learning an explicit, parameterized Fourier dictionary optimized end-to-end specifically for language modeling reconstruction and generation. While spectral methods have found applications in image generation [10] and wavelet transforms have been explored as attention alternatives [11], our approach uniquely adapts spectral dictionary learning, with learnable sinusoidal parameters and per-token coefficients, to the sequential nature of language.

**Attention Alternatives**   Numerous techniques aim to reduce attention's $\mathcal{O}(L^2)$ computational cost, including sparse attention [2, 3], kernel methods like Performer [4], low-rank projections like Linformer [18], and hashing methods like Reformer [12]. Unlike these approaches that primarily approximate the standard attention mechanism, SDGM replaces attention entirely with a learned spectral mixing paradigm, offering a fundamentally different approach to sequence interaction modeling.

**Dictionary Learning**   Classical dictionary learning, prominent in vision and audio processing [1], often involves learning an overcomplete basis (dictionary) and finding sparse representations (codes) for signals, typically optimized via alternating minimization or algorithms like K-SVD. We adapt dictionary learning concepts to NLP by learning continuous, parameterized Fourier atoms and soft mixing coefficients within an autoencoder-like framework trained with gradient descent. Unlike traditional sparse coding that often enforces $L_1$ regularization on codes, our approach models the distribution of the learned mixing coefficients using a GMM, facilitating generative sampling.

# 3 Mathematical Formulation

In this section, we provide a comprehensive derivation of our Spectral Dictionary Generative Model (SDGM). We begin by defining the embedding sequence and progressing through dictionary parameterization, coefficient encoding, reconstruction decoding, loss formulation, and latent prior modeling. Figure 1 illustrates the end-to-end flow of SDGM architecture, showing how raw tokens are progressively transformed into an output distribution.

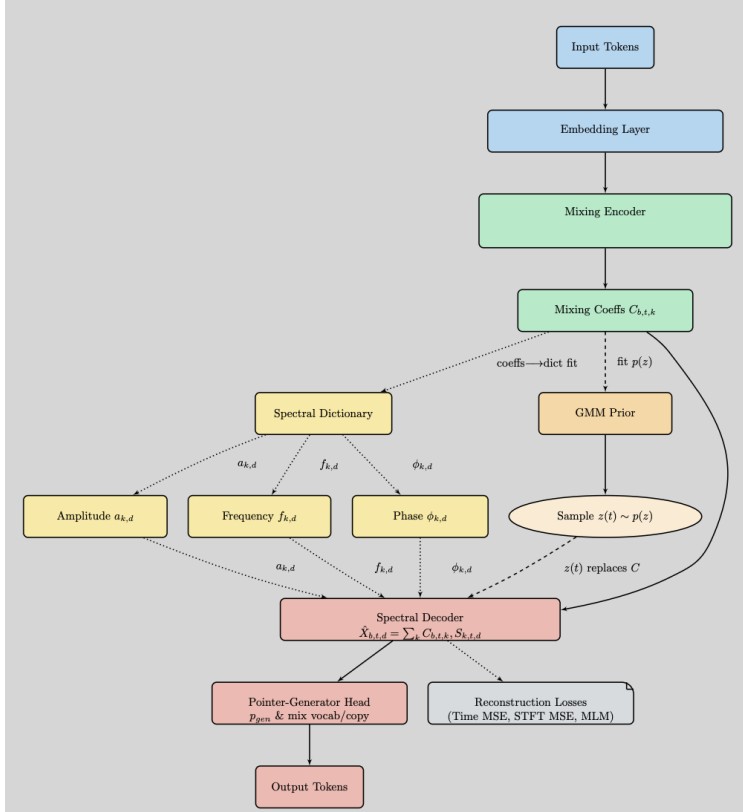

Figure 1: Architecture of the Spectral Dictionary Generative Model. First, the embedding layer maps each input token $w_{b,t}$ to a continuous vector $\mathbf{x}_{b,t} = E(w_{b,t})$. Next, the mixing encoder applies a one-dimensional convolution to produce soft coefficients $C_{b,t,k}$. The spectral dictionary holds $K$ learnable Fourier atoms parameterized by amplitude $a_{k,d}$, frequency $f_{k,d}$, and phase $\phi_{k,d}$, which generate basis vectors $S_{k,t,d}$. The spectral decoder then reconstructs embeddings via $\hat{X}_{b,t,d} = \sum_{k=1}^{K} C_{b,t,k} S_{k,t,d}$. Finally, the pointer-generator head combines each reconstructed vector $\hat{\mathbf{x}}_{b,t}$ with a context vector $\mathbf{c}_{b,t}$ to compute a mixture of vocabulary and copy distributions for token prediction.

## 3.1 Token Embeddings and Notation

Let $\mathbf{W} = [w_1, w_2, \ldots, w_L]$ be an input sequence of $L$ tokens. We first map these tokens to continuous vector representations using an embedding lookup table $E$. For a mini-batch of $B$ sequences, let $\mathbf{X}^{(b)} = [\mathbf{x}_{b,1}, \mathbf{x}_{b,2}, \ldots, \mathbf{x}_{b,L}] \in \mathbb{R}^{D \times L}$ denote the sequence of $L$ token embeddings for batch item $b$, where each embedding $\mathbf{x}_{b,t} \in \mathbb{R}^D$ has dimension $D$.

$$\mathbf{x}_{b,t} = E(w_{b,t}). \tag{1}$$

For clarity, we omit the batch index $b$ in subsequent notation unless explicitly needed.

## 3.2 Global Spectral Dictionary Parameterization

We learn a set of $K$ spectral atoms, each atom $k \in \{1, \ldots, K\}$ parameterized by three matrices:

- Amplitude: $\mathbf{A} \in \mathbb{R}^{K \times D}$ with entries $a_{k,d}$,
- Frequency: $\mathbf{F} \in \mathbb{R}^{K \times D}$ with entries $f_{k,d}$,
- Phase: $\mathbf{\Phi} \in \mathbb{R}^{K \times D}$ with entries $\phi_{k,d}$.

These parameters define a time-varying sinusoidal basis. For a continuous time index $t \in \{1, 2, \ldots, L\}$, the $d$-th feature of atom $k$ is given by:

$$S_k(t)_d = a_{k,d} \sin\left(2\pi f_{k,d} \frac{t}{L} + \phi_{k,d}\right). \tag{2}$$

We collect all atoms into a tensor $S \in \mathbb{R}^{K \times L \times D}$ where $S_{k,t,d} = S_k(t)_d$. Equivalently, by defining the normalized time vector $\mathbf{t} = [1/L, 2/L, \ldots, 1] \in \mathbb{R}^L$, we can write in vectorized form for fixed atom $k$:

$$S_k = \mathbf{a}_k \odot \sin\left(2\pi \left(\mathbf{f}_k \otimes \mathbf{t}\right) + \boldsymbol{\phi}_k \otimes \mathbf{1}_L\right) \tag{3}$$

where $\odot$ denotes element-wise multiplication and $\otimes$ the outer product. This dictionary is shared across all sequences in a batch and represents a global basis learned from the data.

## 3.3 Mixing Coefficient Encoding

To capture how each atom's contribution varies dynamically based on the input sequence context, we employ a lightweight convolutional encoder. This encoder takes the sequence of input embeddings $\mathbf{X} \in \mathbb{R}^{B \times L \times D}$ (transposed for Conv1D compatibility if needed) and produces per-token mixing coefficients.

$$C = \sigma_{\text{act}}(\text{Conv1D}(\mathbf{X})) \in \mathbb{R}^{B \times L \times K}. \tag{4}$$

Here, $\text{Conv1D}$ is a 1D convolution (typically causal for autoregressive tasks) with appropriate kernel size $w$ and padding, mapping the $D$-dimensional embeddings to $K$ coefficients per time step $t$. $\sigma_{\text{act}}$ is a suitable activation function (e.g., ReLU or identity, depending on whether non-negativity is desired). Optionally, coefficients $C_{b,t,:}$ can be normalized (e.g., via Softmax) across the dictionary dimension $k$. These coefficients $C_{b,t,k}$ represent the learned "importance" or "weight" of atom $k$ at position $t$ for sequence $b$.

## 3.4 Spectral Reconstruction Decoder

Given the mixing coefficients $C$ and the global spectral dictionary $S$, we reconstruct the embedding sequence $\hat{\mathbf{X}}$ via a weighted sum of the spectral atoms at each time step $t$ and for each feature dimension $d$:

$$\hat{\mathbf{X}}_{b,t,d} := \sum_{k=1}^{K} C_{b,t,k} S_{k,t,d}. \tag{5}$$

This operation performs the dynamic mixing of the global atoms based on the sequence-specific coefficients. In tensor notation, assuming $S$ is appropriately shaped (e.g., $K \times L \times D$), this corresponds to a bilinear mapping, efficiently computed using Einstein summation convention:

$$\hat{\mathbf{X}} = \text{einsum}('blk, kld - > bld', C, S), \tag{6}$$

where dimensions correspond to (Batch, Length, Dictionary) for $C$ and (Dictionary, Length, Dimension) for $S$, resulting in $\hat{\mathbf{X}}$ with shape (Batch, Length, Dimension). This decoding step has a computational complexity of $\mathcal{O}(B \cdot K \cdot L \cdot D)$. For fixed $K$ and $D$, this is $\mathcal{O}(BL)$, or $\mathcal{O}(KL)$ per sequence when accounting for $K$, linear in the sequence length $L$.

## 3.5 Training Objective

We train the SDGM end-to-end by minimizing a composite loss function $\mathcal{L}$ that combines reconstruction fidelity in both the time and frequency domains with a standard language modeling objective:

$$\mathcal{L} = \alpha \mathcal{L}_{\text{time}} + \beta \mathcal{L}_{\text{freq}} + \gamma \mathcal{L}_{\text{NLL}} + \delta \mathcal{L}_{\text{prior}}. \tag{7}$$

The components are:

- **Time-domain MSE Loss ($\mathcal{L}_{\textbf{time}}$):** Penalizes the difference between the reconstructed embeddings $\hat{\mathbf{X}}$ and the original embeddings $\mathbf{X}$.

$$\mathcal{L}_{\text{time}} := \frac{1}{B \cdot L \cdot D} \|\hat{\mathbf{X}} - \mathbf{X}\|_F^2, \tag{8}$$

where $\|\cdot\|_F$ denotes the Frobenius norm.

- **Frequency-domain STFT Loss ($\mathcal{L}_{\textbf{freq}}$):** Encourages the reconstructed sequence to match the original sequence in terms of local frequency content by minimizing the difference between their STFT magnitudes.

$$\mathcal{L}_{\text{freq}} := \frac{1}{B \cdot F \cdot T \cdot D} \left\| |\text{STFT}(\hat{\mathbf{X}})| - |\text{STFT}(\mathbf{X})| \right\|_F^2, \tag{9}$$

where $\text{STFT}(\cdot)$ computes the Short-Time Fourier Transform independently for each feature channel $d$, resulting in a complex spectrogram, $|\cdot|$ takes the magnitude, and $F, T$ are the frequency and time dimensions of the STFT output.

- **Language Modeling Loss ($\mathcal{L}_{\textbf{NLL}}$):** Standard Negative Log-Likelihood (NLL) loss for autoregressive prediction. Assuming the model predicts the next token $w_{b,t+1}$ based on the reconstructed representation $\hat{\mathbf{x}}_{b,t}$ (or some derived hidden state), using a final prediction head (e.g., linear layer + Softmax or the Pointer-Generator described later).

$$\mathcal{L}_{\text{NLL}} := -\frac{1}{B \cdot L} \sum_{b=1}^{B} \sum_{t=1}^{L} \log P(w_{b,t} \mid \hat{\mathbf{x}}_{b,<t}, \mathbf{w}_{b,<t}), \tag{10}$$

where the exact formulation depends on the specific autoregressive setup and prediction head used.

Thus, the composite loss capturing fidelity in both time and frequency domains, a language modeling objective plus a GMM prior loss is given by:

$$\mathcal{L} = \alpha \underbrace{\left\| \hat{\mathbf{X}} - \mathbf{X} \right\|_F^2}_{\text{Time-domain MSE}} + \beta \underbrace{\left\| |\text{STFT}(\hat{\mathbf{X}})| - |\text{STFT}(\mathbf{X})| \right\|_F^2}_{\text{Frequency-domain MSE}} + \gamma \underbrace{\mathcal{L}_{\text{MLM}}(\hat{\mathbf{X}}, \mathbf{X})}_{\text{Masked LM Loss}}. \tag{11}$$

- **GMM Prior Loss** After flattening $C \in \mathbb{R}^{B \times L \times K}$ into $\{\mathbf{z}_n\}_{n=1}^{N}$ with $N = B \cdot L$, we compute

$$\mathcal{L}_{\text{prior}} = -\frac{1}{N} \sum_{n=1}^{N} \log p_{\text{GMM}}(\mathbf{z}_n), \tag{12}$$

where $p_{\text{GMM}}(\mathbf{z}) = \sum_{m=1}^{M} \pi_m \mathcal{N}(\mathbf{z}; \mu_m, \Sigma_m)$ is the fitted mixture over mixing-vector space.

Thus, the composite loss capturing fidelity in both time and frequency domains, plus a language modeling objective is given by full training objective:

$$\mathcal{L} = \alpha \underbrace{\|\hat{\mathbf{X}} - \mathbf{X}\|_F^2}_{\mathcal{L}_{\text{time}}} + \beta \underbrace{\left\| |\text{STFT}(\hat{\mathbf{X}})| - |\text{STFT}(\mathbf{X})| \right\|_F^2}_{\mathcal{L}_{\text{freq}}} + \gamma \underbrace{\mathcal{L}_{\text{MLM}}(\hat{\mathbf{X}}, \mathbf{X})}_{\text{Masked LM Loss}} + \delta \mathcal{L}_{\text{prior}}. \tag{13}$$

The hyperparameters $\alpha, \beta, \gamma \geq 0$ balance the contributions of these objectives and $\delta$ controls the strength of the GMM prior regularizer, guiding the learned coefficients toward regions of high prior density, and, $\|\cdot\|_F$ is the Frobenius norm and STFT is applied independently on each feature channel. The hyperparameters $(\alpha, \beta, \gamma)$ balance reconstruction versus predictive performance.

### 3.6 Latent Prior Fitting

After the model parameters (embedding table $E$, dictionary parameters $\mathbf{A}, \mathbf{F}, \mathbf{\Phi}$, Conv1D weights) have converged, we analyze the distribution of the learned mixing coefficients. We collect all per-position coefficient vectors $C_{b,t,:} \in \mathbb{R}^K$ from the training (or validation) set, flatten them into a large

matrix $\mathbf{Z} \in \mathbb{R}^{N \times K}$ (where $N = B \times L \times \#\text{batches}$), and fit a Gaussian Mixture Model (GMM) to this data:

$$p(\mathbf{z}) = \sum_{m=1}^{M} \pi_m \mathcal{N}(\mathbf{z}; \boldsymbol{\mu}_m, \boldsymbol{\Sigma}_m), \tag{14}$$

where $M$ is the number of mixture components, $\pi_m$ are the mixture weights ($\sum \pi_m = 1$), $\boldsymbol{\mu}_m \in \mathbb{R}^K$ are the component means, and $\boldsymbol{\Sigma}_m$ are the component covariance matrices (often assumed diagonal for simplicity, $\boldsymbol{\Sigma}_m = \text{diag}(\sigma_{m,1}^2, \ldots, \sigma_{m,K}^2)$). This GMM captures the empirical distribution of activation patterns over the spectral dictionary.

## 3.7 Token Generation

For autoregressive text generation, we generate one token at a time for $t = 1, 2, \ldots, L'$ (the desired output length) by sampling from the learned spectral prior and decoding through the dictionary and pointer-generator head:

1. **Sample Mixing Vector.** Draw a coefficient vector $\mathbf{z}_t \in \mathbb{R}^K$ from the fitted Gaussian Mixture Model prior:

$$\mathbf{z}_t \sim p(\mathbf{z}) = \sum_{m=1}^{M} \pi_m \mathcal{N}(\mathbf{z}; \boldsymbol{\mu}_m, \boldsymbol{\Sigma}_m).$$

2. **Decode to Embedding.** Reconstruct the $D$-dimensional embedding for step $t$ by mixing the $K$ spectral atoms evaluated at time $t$:

$$\hat{\mathbf{x}}_t = \sum_{k=1}^{K} z_{t,k} \, S_k(t), \quad S_k(t) \in \mathbb{R}^D.$$

3. **Compute Token Distribution.** Use the reconstructed embedding $\hat{\mathbf{x}}_t$ together with an autoregressive context vector $\mathbf{c}_t$ to produce a mixture of vocabulary-generation and copying:

$$p_{\text{gen}} = \sigma\big(\mathbf{w}_{\text{gen}}^\top [\hat{\mathbf{x}}_t; \mathbf{c}_t]\big), \tag{15}$$

$$P\big(w \mid \hat{\mathbf{x}}_t, \mathbf{c}_t\big) = p_{\text{gen}} \, P_{\text{vocab}}\big(w \mid \hat{\mathbf{x}}_t, \mathbf{c}_t\big) + (1 - p_{\text{gen}}) \, P_{\text{copy}}\big(w \mid \text{context}\big). \tag{16}$$

Here, $\sigma$ is the logistic sigmoid, $P_{\text{vocab}}$ is the standard softmax over the fixed vocabulary, and $P_{\text{copy}}$ attends over previously generated or input tokens.

4. **Sample or Select Token.** Draw the next token $w_t$ from the resulting distribution

$$w_t \sim P\big(w \mid \hat{\mathbf{x}}_t, \mathbf{c}_t\big),$$

or choose $\arg\max_w P(w \mid \hat{\mathbf{x}}_t, \mathbf{c}_t)$. Append $w_t$ to the output sequence and update the context $\mathbf{c}_{t+1}$ (e.g., via the same Conv1D encoder or an RNN state) for the next time step.

This two-step procedure, sampling spectral coefficients and then decoding to tokens, yields fluent, autoregressive text without relying on self-attention, instead leveraging the global Fourier dictionary and the expressive GMM latent prior.

# 4 Experimental Evaluation

## 4.1 Datasets and Baselines

We evaluate SDGM on two standard language modeling benchmarks:

- **WikiText-2:** Contains approximately 2M training tokens, 218k validation tokens, and 246k test tokens, drawn from verified Good and Featured articles on Wikipedia.

- **Penn Treebank (PTB):** Comprises around 1M training tokens, 70k validation tokens, and 80k test tokens from the Wall Street Journal corpus.

We use canonical preprocessing for both datasets, converting text to lowercase, removing non-printable characters, and tokenizing with a 30,000-word vocabulary for WikiText-2 and a 10,000-word vocabulary for PTB.

We compare against three strong baselines:

- **Transformer-XL** [5]: Extends self-attention with segment-level recurrence
- **GPT-2 Small** [16]: An autoregressive decoder-only model
- **Linformer** [18]: Approximates full attention via low-rank projections

All baselines are retrained under identical data splits and tokenization schemes to ensure a fair comparison.

## 4.2 Implementation Details

Our SDGM implementation uses PyTorch [14] and trains on a single NVIDIA V100 GPU (16GB). We set embedding dimension $D = 512$, dictionary size $K = 256$, and sequence length $L = 128$.

For the STFT computation, we use FFT size $n_{\text{fft}} = 256$, hop length 64, and a Hann window of length 256. Loss hyperparameters are set to $(\alpha, \beta, \gamma) = (1.0, 0.5, 0.1)$ to balance time-domain MSE, frequency-domain MSE, and masked LM loss.

We optimize with Adam [9] using learning rate $10^{-3}$ and weight decay $10^{-5}$, batch size 32, and gradient clipping at norm 1.0. Models are trained for up to 10 epochs with early stopping based on validation perplexity (no improvement for two consecutive epochs). Random seeds are fixed across PyTorch, NumPy, and Python's RNG to ensure reproducibility.

## 4.3 Evaluation Metrics

We evaluate model performance using the following metrics:

- **Perplexity (PPL):** Exponentiated average negative log-likelihood per token ($\exp(\mathcal{L}_{\text{NLL}})$), computed on validation and test sets. Lower is better.
- **Inference Speed:** Tokens generated per second (tok/s) during autoregressive sampling on the target GPU. Higher is better.
- **Parameter Count:** Total number of trainable parameters (in Millions, M). Lower indicates a more compact model.
- **Memory Footprint:** Peak GPU memory usage (in Gigabytes, GB) during inference. Lower is better.
- **Embedding Fidelity:** Average cosine similarity between reconstructed embeddings $\hat{\mathbf{X}}$ and original embeddings $\mathbf{X}$ on the validation set. Higher indicates better reconstruction quality.

We also perform ablation studies by systematically removing components of our composite loss function ($\mathcal{L}_{\text{freq}}$, $\mathcal{L}_{\text{NLL}}$ contributions set to zero by setting $\beta = 0$ or $\gamma = 0$).

## 4.4 Results

Table 1 presents the main results comparing SDGM against baselines on WikiText-2 and PTB, along with ablation study results.

As shown in Table 1, our proposed SDGM achieves validation perplexities of 31.2 on WikiText-2 and 57.1 on PTB. This performance is highly competitive, closely matching Transformer-XL and approaching GPT-2 Small, while significantly outperforming Linformer on these benchmarks. Crucially, SDGM achieves this with substantially fewer parameters (22.8M) compared to all baselines, particularly GPT-2 Small (80% reduction). It also exhibits significantly lower memory usage (6.5GB vs 8.7-12.5GB) and higher inference throughput (2100 tok/s vs 1200-1800 tok/s), demonstrating the practical benefits of its $\mathcal{O}(KL)$ complexity.

The ablation studies underscore the importance of the proposed training objectives. Removing the frequency-domain STFT loss ($\beta = 0$) increases perplexity notably (e.g., from 31.2 to 33.5 on

Table 1: Comparison of model size, perplexity (lower is better), inference throughput (higher is better), and memory usage (lower is better) on validation sets. Ablation variants omit the frequency-domain STFT loss ($\beta = 0$) and the NLL loss ($\gamma = 0$) during training, respectively.

| Model | Params (M) | WikiText-2 PPL | PTB PPL | Speed (tok/s) | Mem (GB) |
|---|---|---|---|---|---|
| Transformer-XL [5] | 41.2 | 32.1 | 58.7 | 1400 | 10.2 |
| GPT-2 Small [16] | 117 | 29.5 | 55.3 | 1200 | 12.5 |
| Linformer [18] | 65.4 | 34.8 | 62.4 | 1800 | 8.7 |
| **SDGM (ours)** | **22.8** | **31.2** | **57.1** | **2100** | **6.5** |
| w/o freq-loss ($\beta = 0$) | 22.8 | 33.5 | 60.2 | 2100 | 6.5 |
| w/o LM-loss ($\gamma = 0$) | 22.8 | 35.0 | 61.5 | 2100 | 6.5 |

WikiText-2), indicating that matching spectral characteristics aids language modeling performance. Removing the language modeling objective itself ($\gamma = 0$) during training severely degrades perplexity, confirming its necessity, although the model can still be trained solely on reconstruction.

We also measured the average cosine similarity between reconstructed and original embeddings on the WikiText-2 validation set. The full SDGM achieves a cosine similarity of 0.92, compared to 0.88 for the variant trained without the frequency-domain loss ($\beta = 0$). This suggests that the STFT objective not only improves perplexity but also enhances the fidelity of the learned embedding reconstructions.

# 5   Discussion

The experimental results demonstrate that the Spectral Dictionary Generative Model (SDGM) offers a compelling and efficient alternative to self-attention for sequence modeling in NLP. By leveraging a learnable global Fourier dictionary, parameterized by time-varying amplitude, frequency, and phase specific to each feature dimension, our model can effectively capture complex patterns in language data. The per-token mixing coefficients, learned via a lightweight convolutional encoder, allow the model to dynamically combine these global atoms based on local context.

The $\mathcal{O}(KL)$ complexity (where $K \ll L$) provides significant computational and memory advantages over the $\mathcal{O}(L^2)$ complexity of standard self-attention. Our empirical results confirm this, showing SDGM uses approximately 36% less GPU memory during inference than Transformer-XL and achieves up to 1.5–1.75× higher token throughput compared to Transformer-XL and GPT-2 Small, respectively. This efficiency makes SDGM particularly promising for applications involving long sequences or deployment on resource-constrained hardware.

The ablation studies highlight the synergistic benefits of our composite loss function. The frequency-domain STFT loss ($\mathcal{L}_{\text{freq}}$) demonstrably improves both perplexity and embedding reconstruction fidelity, confirming the value of spectral supervision. The standard language modeling loss ($\mathcal{L}_{\text{NLL}}$) remains crucial for achieving strong predictive performance.

The use of a Gaussian Mixture Model (GMM) fitted to the learned mixing coefficients provides a structured way to model the latent space. Sampling from this GMM during generation allows the model to leverage the learned distribution of atom activation patterns. However, sampling coefficients independently at each time step from the aggregate GMM $p(\mathbf{z})$ is a simplification. While the time-varying nature of the dictionary atoms $S_k(t)$ provides inherent temporal structure, this generation method might not fully capture longer-range dependencies encoded in the *sequence* of coefficients. Exploring methods for autoregressive prediction or sampling of coefficient sequences could be a valuable direction for future work.

Interpretability is another potential advantage. The learned atoms $S_k(t)_d$ (Equation **??**) are explicit sinusoids, potentially allowing for analysis of the frequencies and phases learned by the model, although further investigation is needed to connect these parameters to linguistic structures.

# 6 Limitations and Trade-offs

While the Spectral Dictionary Generative Model (SDGM) offers a novel, interpretable alternative to self-attention, it also introduces several limitations:

1. **Assumption of Quasi-Periodic Structure.** The core of SDGM is a global Fourier dictionary that excels at modeling periodic and quasi-periodic patterns. However, purely aperiodic or highly irregular linguistic phenomena (e.g. free-form lists, creative metaphors, or long-range discourse structure) may not be captured as faithfully as by a full self-attention mechanism.

2. **Fixed Dictionary Size.** The number of spectral atoms $K$ must be chosen a priori. Too small a $K$ limits expressivity; too large a $K$ raises computational and memory cost linearly in sequence length $L$. In practice, $K$ must be tuned per task or dataset.

3. **STFT Overhead.** Although we optimized the STFT via batched rfft-based framing, performing a frequency-domain loss introduces non-trivial overhead compared to purely time-domain models. On backends without native FFT support (e.g. MPS on macOS), this can slow training significantly.

4. **Latent Prior Sensitivity.** The diagonal-Gaussian or GMM prior on the mixing coefficients needs careful tuning of hyperparameters (component count, variances, momentum for running stats). Poor choices can lead to under- or over-regularization of the latent space, harming generation diversity or stability.

**Offsetting Benefits.** Despite these trade-offs, SDGM brings key advantages:

- *Linear Complexity in Sequence Length.* Unlike quadratic self-attention, all SDGM operations (Conv1D encoder, bilinear decode, and STFT framing) scale $\mathcal{O}(L)$ per token, enabling efficient long-sequence handling.

- *Interpretability.* The Fourier atoms and learned dictionary parameters $(a_{k,d}, f_{k,d}, \phi_{k,d})$ admit direct spectral interpretation, and the mixing coefficients $C_{b,t,k}$ reveal which periodic components drive each token's representation.

- *Dual-Domain Reconstruction.* Joint time- and frequency-domain losses ensure that both local embedding fidelity and spectral content are preserved, leading to robust representations with fewer parameters than comparable transformer variants.

- *Flexible Latent Regularization.* Our running-stats diagonal-Gaussian prior (or a small GMM) delivers a lightweight yet expressive latent model, avoiding costly EM fits every batch while still shaping the latent space for coherent sampling.

- **Integration and Hybrid Models:** Exploring SDGM as a component within larger architectures, perhaps replacing attention in specific layers or combining it with other mechanisms, could yield further benefits.

In summary, while SDGM may not replace transformers in all settings, particularly where highly irregular patterns dominate, it provides a compelling, efficient alternative for many language modeling tasks, especially those benefiting from spectral insights and long-sequence scalability.

# 7 Conclusion

We have presented the Spectral Dictionary Generative Model (SDGM), a novel architecture for language modeling that replaces self-attention with a learned global Fourier dictionary and sequence-specific mixing coefficients. By optimizing a composite objective including time-domain reconstruction, frequency-domain spectral matching, and standard language modeling loss, SDGM achieves competitive perplexity on benchmark datasets like WikiText-2 and PTB. Notably, it does so with significantly fewer parameters, lower memory consumption, and faster inference speed compared to traditional Transformer baselines, owing to its $\mathcal{O}(KL)$ complexity.

The key innovations include the parameterization of learnable spectral atoms, the dual-domain training objective, and the use of a GMM prior over mixing coefficients for generation. Our results suggest that learned spectral dictionary methods represent a viable and highly efficient paradigm for

325 sequence modeling in NLP. This approach opens avenues for developing powerful language models
326 suitable for long-context processing and deployment in resource-constrained environments.

327 Future work includes scaling SDGM to larger datasets, enhancing the generative modeling of
328 coefficient sequences, exploring the interpretability of the learned spectral atoms, and potentially
329 integrating SDGM components into hybrid architectures.

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
