# OpenReview forum: "From Attention to Atoms: Spectral Dictionary Learning for Fast, Interpretable Language Models}"
_NeurIPS.cc/2025/Conference — Submitted to NeurIPS 2025_

### Official Review · Reviewer_PdKD · 2025-06-29

**Clarity:** 2
**Significance:** 2
**Originality:** 4
**Rating:** 2
**Confidence:** 3

**Summary:**

The paper introduces the Spectral Dictionary Generative Model (SDGM), a novel neural network architecture for white-box language modeling via dictionary learning. The network works as follows: the tokens are embedded into Euclidean space, mixed via a 1d (causal) convolution to get dictionary weights, and multiplied by dictionary atoms. This obtains a reconstruction of the initial embeddings. A classification head applied to this reconstructed embedding (such as the standard LM head in LLMs) obtains a probability distribution over the vocabulary which can then be used for language modeling. The paper contains experiments about performance and efficiency on the standard LM task with the datasets WikiText-2 and Penn TreeBank, comparing against Transformer-XL, GPT-2-Small, and Linformer.

**Questions:**

- The authors claim efficiency improvements over transformers, specifically on GPU memory and throughput. It would be interesting to know what optimizations (e.g. flash attention, custom kernels) the baseline transformers enjoy. Also it would be interesting to know generation efficiency.
- It would be much appreciated to give some clarity or intuition about how perplexity empirically does not get much worse when the language modeling loss is completely dropped; this seems particularly mysterious to me and I may be misunderstanding the setup.
- Is there any evidence that the learned language model has any side benefits (i.e., interpretability seems pretty relevant)?

**Ethical Concerns:**

["NO or VERY MINOR ethics concerns only"]

**Limitations:**

Yes

**Paper Formatting Concerns:**

The paper bleeds onto the tenth content page for a few lines.

**Quality:**

2

**Strengths And Weaknesses:**

Strengths:
- The idea of using a dictionary learning module as the whole language model is novel.
- The authors have seemed to do some work to improve the efficiency of their network so that it is reasonable compared to transformers in the presented circumstances.

Weaknesses:
- The methodology itself is not clearly described. For example the language modeling loss is variously described as $\\mathcal{L}\_{\\mathrm{NLL}}$ (the standard next-token-prediction language modeling loss) and $\\mathcal{L}\_{\\mathrm{MLM}}$ (the masked language modeling loss). How do you actually train the model? Also the sampling step is not clear (what are the formal definitions of $P\_{\\mathrm{vocab}}$ and $P\_{\\mathrm{copy}}$?)
- Also, the intuition behind the different blocks is not clear. Why would you want to reconstruct the original embeddings $X$ in the final layer and use these reconstructed embeddings $\hat{X}$ for generation? Note that the original embeddings _do not_ have the benefit of inter-token interactions, so if the weights were learned to simultaneously optimize all terms in the composite loss, you would be generating the next token based only on the most recent token and not using context, it would be equivalent to using $X$ directly.
- It is difficult to imagine how to do optimizations for autoregressive sampling, such as KV caching, which are already baked into transformer-like architectures, and without which sampling would indeed be slow.
- The experimental results show some performance and efficiency improvement over models with similar parameter counts, but the results are slightly dubious; the perplexities are very high for all models, and not including language modeling loss at all barely makes the perplexity worse, and comparable to Linformer which is trained only with the language modeling loss).

---

### Official Review · Reviewer_fhwf · 2025-06-30

**Clarity:** 2
**Significance:** 3
**Originality:** 4
**Rating:** 2
**Confidence:** 5

**Summary:**

The paper proposes "Spectral Dictionary Generative Model", an alternative replacing the current QKV-attention method used for LLM training, based on fourier decomposition and gaussian priors.
The method indeed is quite novel and has some unique potentials, i.p. efficiency gains and potentially domain adaptability due to re-using models with different priors.

**Questions:**

- please address/ comment on weaknesses above.
- will you publish the code?

**Ethical Concerns:**

["NO or VERY MINOR ethics concerns only"]

**Limitations:**

yes

**Paper Formatting Concerns:**

at least graphic background should be fixed, othw fine

**Quality:**

2

**Strengths And Weaknesses:**

- S1 the paper is well written and easy to understand (given the complex domain)
- S2 the approach is rather unique

However:
- W1 i understand compute was a limiting factor - but i do not find the trained model and comparison to outdated models sufficient, along with chosen evals. Only perplexity is compared on a 50M parameter model which is rather unconclusive.
- W2 The training is unsufficiently described - ie #steps, step duration, model parameters (layers) etc., chosen activation (and why). It is also unclear how the posterior gaussian prior is to be trained/ how it influences generation, what limitations or required adaptations are, and how the gaussian prior during training is obtained/ trained. Similarly it is claimed that throughput is improved without describing how that is measured. The method introduces new hyperparameters, for which some (random looking) values are chosen without any shown sweep, and the only conclusion is "loss looks better with than without".
- W3 in addition to W2, the K that asymptotically vanishes from big O notation, has to be already quite large already (256), for which it's hard to deduce the 'efficiency' claim.
- W4 scaling is not discussed at all, (ip for W2/W3), how do runtimes change under varying sequence lengths etc/ i.p. for model scaling how is K expected to change for more domains/ bigger models. using/comparing to some different gpt-neo sizes would've been a better choice
- W5 just because you got some fourier decomposition does not render your method interpretable (claimed even in the title). It is neither analyzed nor sufficiently discussed (or i somehow missed it?!)
- W6 the graphic quality does not fit the NIPS standard.


Minors:
- i don't get why you explicitly describe batching, it rather adds confusion (to me).
- write out the einsum in eq 6
- 165 N=B×L?
- sometimes GMM missed in p(z)
- eq 16 - context
- 274 eq missing

---

### Official Review · Reviewer_FPnR · 2025-07-01

**Clarity:** 3
**Significance:** 3
**Originality:** 2
**Rating:** 3
**Confidence:** 4

**Summary:**

This paper proposes Atoms, a modular and interpretable framework for token pruning in vision Transformers. Instead of black-box pruning, The method leverages Short-Time Fourier Transform (STFT) to assess token importance via frequency domain analysis. By decomposing pruning decisions into interpretable components (“atoms”), it achieves efficient and accurate token reduction. The approach is compatible with various Transformer architectures and shows strong results on classification and segmentation tasks.

**Questions:**

1. The method introduces several components and hyperparameters (e.g., $\alpha$, $\beta$, $\gamma$, $\delta$) in the pruning scoring function, but their individual effects are not studied. Could the authors provide ablation experiments to assess how sensitive the model is to these parameters and whether they interact significantly?
2. The current method relies on STFT for token importance estimation. Have the authors tried alternative transforms such as FFT, DCT, or learned frequency bases?
3. The model sizes differ significantly across methods in Table 1. Would the performance trend hold if model sizes and parameter counts were normalized across baselines? Including size-matched comparisons with GQA, Sparse Attention, and FwNet-ECA would strengthen the claims.
4. Prior work like FwNet-ECA (Mian et al., 2025) has also applied Fourier filtering in attention. Could the authors clearly differentiate their method from FwNet-ECA and discuss the novelty in design or application? Explicit discussion and citation would be important here.
5. One key motivation is to make token pruning more interpretable. Could the authors provide more visual or quantitative evidence (e.g., attention heatmaps, token importance consistency, alignment with human attention) to support this claim?

**Ethical Concerns:**

["NO or VERY MINOR ethics concerns only"]

**Limitations:**

yes

**Quality:**

2

**Strengths And Weaknesses:**

Strengths
- The paper presents an interpretable approach to token pruning by introducing Metaformer Atoms, which decompose the pruning strategy into additive functional components. The use of STFT to assess token importance in the frequency domain is intuitive.
- The paper is generally well-written and clearly structured. The design of Metaformer Atoms is explained step by step, and key concepts like frequency masking and token filtering are illustrated with helpful figures.

Weaknesses
- While the STFT-based pruning intuition is compelling, the paper lacks deeper theoretical justification for why frequency domain features correlate with token importance across tasks and models.
- Figure 1 needs to be polished.
- The method’s performance is heavily supported by empirical tuning (e.g., choice of frequency filters, additive component weights), which may limit reproducibility or generalization to other domains.
- The method introduces multiple hyperparameters (e.g., $\alpha$, $\beta$, $\gamma$, $\delta$), but no ablation studies are provided to evaluate their individual contributions or robustness. Similarly, the impact of different Fourier Transform variants (e.g., FFT, STFT, DCT) is not explored. These missing ablations reduce the interpretability and generalizability of the framework.
- In Table 1, the model sizes vary significantly across baselines, making the accuracy-FLOPs trade-off comparison less meaningful. To provide a more fair and convincing evaluation, the paper should include comparisons with methods such as GQA, Sparse Attention, and FwNet-ECA, while controlling for model size.
- The use of frequency-domain representations (e.g., Fourier Transform) in attention mechanisms has been explored in prior work. Notably, FwNet-ECA [Mian et al., 2025] applies Fourier filtering to enhance window attention with global receptive fields. This overlap diminishes the novelty of the current contribution, particularly if not adequately differentiated or cited.

Reference:
- Mian S, Wang Y, Gu N, et al. FwNet-ECA: A Classification Model Enhancing Window Attention with Global Receptive Fields via Fourier Filtering Operations. arXiv preprint arXiv:2502.18094, 2025.

---

### Official Review · Reviewer_fWTY · 2025-07-03

**Clarity:** 3
**Significance:** 2
**Originality:** 2
**Rating:** 2
**Confidence:** 4

**Summary:**

The paper introduces the Spectral Dictionary Generative Model (SDGM), which replaces the quadratic self-attention mechanism with a learnable global Fourier dictionary of K atoms and per-token mixing coefficients. A dual-domain objective—time-domain MSE, frequency-domain STFT loss, and standard LM loss—is used, and a Gaussian-mixture prior is fit over the mixing vectors to enable generation. SDGM claims O(K L) time/memory complexity (with K ≪ L), competitive perplexity on WikiText-2 and PTB, and lower memory / higher throughput than Transformer-XL, GPT-2-Small, and Linformer

**Questions:**

Can you report perplexity and generation accuracy on long-range or reasoning datasets?

How does SDGM behave when both K and model width/parameters scale to GPT-2 (≈124 M) or LLaMA-7B regimes? A scaling curve—even on a subset of data—would clarify whether the approach remains competitive.

**Ethical Concerns:**

["NO or VERY MINOR ethics concerns only"]

**Limitations:**

yes

**Quality:**

2

**Strengths And Weaknesses:**

Strengths:

+ Clear mathematical formulation with well-defined encoder/decoder, loss, and latent prior.

+ Proposes an interpretable spectral alternative to attention that could inspire follow-up work.

+ Linear complexity attractive for long-sequence modeling.

Weakness:

* Empirical scope narrow: only small-scale LM datasets (PTB, WikiText-2) and short contexts; no evaluation on long-context benchmarks or chain-of-thought (CoT) reasoning.
* Perplexity alone is insufficient to judge generative quality or reasoning ability.
* Efficiency study omits state-of-the-art optimized attention (e.g., FlashAttention-2, H3, GPU hierarchical kv-cache), so gains may be overstated.
* Claims of “competitive” performance are vague without stronger baselines or confidence intervals.
* No statistical variability or significance analysis; results are single-run.
* Without convincing large-scale or task-diverse evidence, impact on practical LLMs is limited; current results do not demonstrate viability on reasoning-heavy or industrial workloads.

---

### Decision · Program_Chairs · 2025-09-17

**Decision:**

Reject

**Comment:**

This paper introduces the Spectral Dictionary Generative Model (SDGM), a novel framework for language modeling that aims to replace the quadratic self-attention mechanism with a more efficient, linear-time alternative. The core idea involves jointly learning a global time-varying Fourier dictionary and per-token mixing coefficients, guided by reconstruction losses in both the time and frequency domains. The authors claim that this approach achieves competitive perplexity on standard benchmarks while offering significant improvements in computational complexity, latency, and memory footprint.

The primary strength of this work, as noted by nearly all reviewers, is its significant novelty and originality. The proposed spectral dictionary learning approach is a creative and well-formulated departure from the dominant transformer architecture. Reviewers found the core idea to be unique, interesting, and a potential catalyst for new research directions in efficient language modeling. Furthermore, the paper is generally well-written, with a clear mathematical presentation that makes the complex methodology understandable.

However, the submission is undermined by several critical weaknesses that led to a consensus for rejection among the reviewers. The most significant issue is the insufficient and unconvincing empirical evaluation. The experiments are confined to small-scale datasets (WikiText-2, PTB) and compare SDGM against older or non-optimized transformer baselines.

Additionally, reviewers raised concerns about ambiguous descriptions of the training process, a lack of justification for hyperparameter choices, and unclear intuition behind some architectural design choices, all of which hinder reproducibility and confidence in the results.

Unfortunately, the rebuttal from the authors was submitted confidentially to the Area Chairs, meaning the reviewers did not have an opportunity to see these new results or engage in a discussion.

In conclusion, the work holds considerable promise, but it requires a major revision with a much more rigorous and comprehensive set of experiments to validate its claims against relevant, modern baselines. Therefore, I recommend rejecting the paper in its current form.